# Assessment of the Potential Ability to Penetrate into the Hard Tissues of the Root of an Experimental Preparation with the Characteristics of a Dental Infiltratant, Enriched with an Antimicrobial Component—Preliminary Study

**DOI:** 10.3390/ma14195654

**Published:** 2021-09-28

**Authors:** Małgorzata Fischer, Małgorzata Skucha-Nowak, Bartosz Chmiela, Anna Korytkowska-Wałach

**Affiliations:** 1Unit of Dental Propedeutics, Department of Conservative Dentistry with Endodontics, Faculty of Medical Sciences in Zabrze, Medical University of Silesia, 40-055 Katowice, Poland; mskucha-nowak@sum.edu.pl; 2Department of Materials Technologies, Faculty of Materials Engineering, Silesian University of Technology, 40-019 Katowice, Poland; bartosz.chmiela@polsl.pl; 3Department of Organic Chemistry, Bioorganic Chemistry and Biotechnology, Faculty of Chemistry, Silesian University of Technology, 44-100 Gliwice, Poland; anna.korytkowska-walach@polsl.pl

**Keywords:** root caries, infiltration, microstomatology, dental materials, antimicrobial properties, polymers

## Abstract

Infiltration is a method of penetration with a low viscosity resin that penetrates deep into demineralised tooth tissue and fills the intergranular spaces, hence reducing porosity. Carious lesions initially located at the enamel–cement junction are usually found in elderly patients. Those spots are predisposed to bacterial adhesion originating both from biofilm and from gingival pocket bacteria. The aim of this study was to evaluate the penetration of an experimental preparation, which has the characteristics of a dental infiltrant, enriched with an antibacterial component, into the decalcified root cement tissues of extracted human teeth in elderly patients. An experimental preparation with the characteristics of a dental infiltrant was prepared, applied, and polymerised on the surface of extracted, previously decalcified human teeth. The control sample was Icon (DMG, Hamburg, Germany). The ability of the preparations to penetrate deep into the root cement was evaluated using scanning electron and light microscopy. The study showed that an experimental preparation could potentially be used for treatment of early carious lesions within the tooth root in elderly patients, among others, as it penetrates deep into demineralised tissues. More research is needed.

## 1. Introduction

Despite extensive development ongoing in medicine, dental caries is still one of the most common diseases of the oral cavity. As a result, demineralization and proteolytic tissue breakdown occur. The carious process involves both tissues located in the coronal part, such as enamel and dentin, and those located in the root part, such as root cementum and root dentin [1,2,3,4,5].

Microinvasive dentistry (also known as microstomatology) is a modern concept based on early detection of carious lesions at the molecular level. This new branch in restorative dentistry makes it possible to stop the development of the disease quickly and to restore the normal structure and function of tooth tissue [6,7,8,9]. In order to assess the degree of demineralisation, diagnostic equipment characterised by a wide range of sensitivity and high specificity was used [10,11,12,13,14,15]. 

One of the basic principles of microstomatology is the remineralisation of hard dental tissues. Preparations used for that purpose contain fluoride compounds, such as sodium fluoride, acidulated phosphate fluoride (APF), tin fluoride, and amine fluoride [16,17,18]. Fluoridisation strengthens tooth enamel against the action of bacteria and acids metabolised by them, reduces tooth hypersensitivity in the cervical region, and prevents the formation of carious lesions in areas with previously placed fillings. However, fluoridation preparations have disadvantages. Fluoride compounds applied to tooth surfaces are not mechanically resistant, especially during oral hygiene procedures. These preparations are also not able to penetrate deep into demineralised tooth tissue. Another disadvantage is the lack of ability to inhibit the development of microorganisms on their surface, which would facilitate long-term inhibition of carious lesion development. Moreover, allergy to any of its ingredients is also a contraindication for the use of fluoride preparations. 

Infiltration is an answer to the disadvantages of fluoride preparations—it is also a method that does not require surgical interference in the hard tissues of the tooth. This method comprises deep penetration into the demineralised tooth tissues, while filling the intercrystalline spaces with low viscosity resin and reducing the tissue porosity. As a result, the window for penetration of bacterial toxins into the hard tissues and the bacteria producing them is closed. The result is inhibition of the decalcification process and caries development [19,20]. An infiltrant is a substance based on polymeric resins, capable of penetrating tissues due to capillary forces. According to Manji [21,22], dental infiltrants are used to treat lesions on smooth vestibular and tangential surfaces at the white spot stage, with a maximum radiological depth of up to 1/3 of the outer dentin layer. The advantages of that method include mechanical stabilisation of demineralised tissue, permanent closure of superficial micropores, and inhibition of the progression of lesions. In this way, we minimise the risk of secondary caries and delay surgical intervention, while at the same time achieving aesthetic benefits [23,24]. The use of infiltrants also reduces the risk of gingivitis and the occurrence of tooth hypersensitivity in the cervical region [25]. The requirements for infiltrants to be used in treatment of carious lesions include: a high surface tension, a low density, hydrophilicity, a lack of toxicity to the host, a resistance to chemical and mechanical effects, the ability to polymerise to a solid state, the similarity of the infiltrant colour to that of the tooth, a lack of interaction with food and drugs, and the ability to inhibit multiplication of microorganisms on their surface. The last feature mentioned is very important as it reduces the risk of secondary caries. The only commercially available preparation of such kind is Icon (DMG, Hamburg, Germany), launched in 2009. It is used for the treatment of early enamel caries lesions located on the vestibular and proximal surfaces. However, this preparation does not meet all the requirements for infiltrants, as it does not have a component responsible for bacteriostaticity [25]. This fact prompted the authors of the study to undertake research towards the synthesis of an experimental preparation enriched with the missing component. The developed experimental preparation was enriched with a PMMAn-MTZ monomer, which has adhesive properties. Metronidazole acts on Gram-negative anaerobic microorganisms and Gram-positive anaerobic bacteria (organisms living under anaerobic or low oxygen conditions). Synthesis of PMMAn with metronidazole provides a component that potentially inhibits microbial proliferation on the surface of the hard tooth tissue. PMMAn and metronidazole are linked by an ester bond, which is hydrolysed when exposed to water (saliva). In this way, metronidazole can interact with the experimental preparation to inhibit bacterial growth on the tooth surface continuously [26]. According to the instructions of the manufacturer, the commercial preparation Icon is mainly used for caries within the enamel; therefore, in the opinion of the authors of this study, the potential of infiltrants is not fully exploited. An attempt was made to apply the synthesised infiltrant to the surface of decalcified root cement.

Root cement is the mineralized hard tissue of the tooth that covers the outer surface of the tooth root. Thickness of root cement is locally variable, being thickest in the apical region (150–200 micrometres) and thinnest in the cervical region (20–50 micrometres) [27,28].

Carious lesions initially located at the enamel–cement junction are usually found in older patients [29,30]. As life expectancy increases, there is no apparent correlation with the number of retained healthy teeth [27]. Modern dentistry faces the challenge of improving oral health. With age, there is an increase in the number of general co-morbidities and medications taken by patients, as well as reduced manual dexterity [31]. All these factors contribute to the development of root caries. In the population, we also observed generalized senile periodontal atrophy, characterized by bone loss of the alveolar process and the alveolar part of the mandible, as well as gingival recession. 

As a result, the clinical crowns of the teeth are elongated, while the roots and cementum are exposed [32]. The root surface exposed to the oral environment is much more sensitive to mechanical and chemical destruction than the highly mineralised enamel. This process is initially asymptomatic, manifesting itself as a small brown demarcated lesion (grade 1 according to Billings) below a well-mineralised surface layer [33]. Demineralisation of the root cement is similar to that observed in enamel (destruction of apatite crystals occurs subsurface, before bacterial penetration) [29]. Some differences are also apparent in caries of enamel and root cement. Demineralisation in root cement starts much faster than in enamel, already at the plaque pH of 6–6.5 [34]. In addition, the root surface contains less fluoride than the superficial layer of enamel, which is related to its lower exposure to fluoride contained in saliva and preparations used to maintain oral hygiene [35]. Bacterial invasion occurs at an earlier stage of root caries than in case of enamel caries. The accumulation of bacteria at the cemento–dentin interface causes rapid decomposition of collagen fibres. The presence of traumatic nodes in the oral cavity is not without significance in the formation of decalcifications in the cervical region in case of elderly patients. Bite loads acting eccentrically to the long axis of the tooth cause a loss of connection between hydroxyapatite crystals, which subsequently leads to their disruption and the formation of tissue loss in the cervical region [36]. An oral environment rich in carious bacteria and poor in saliva adversely affects the structure of the exposed cement, leading to the formation of carious defects. 

There are few reports in the available literature on the use of infiltrants enriched with an antibacterial component [37]. The authors of the study focused on the potential attempt to apply infiltrants on the surface of the root cement in order to achieve the intended therapeutic effect that is minimally invasive for the hard tissues of the tooth. 

The aim of this study was to assess the potential degree of penetration of the experimental preparation valuate, which reached deep into the decalcified root cement tissues of extracted human teeth, with the characteristics of a dental infiltrant enriched with an antibacterial component in adult patients, including the elderly. 

This study is a pioneering and pilot study in this field. The authors based their work on the assessment of the degree of penetration of the experimental preparation based on the SEM observations. 

## 2. Materials and Methods

The material for the study included 12 extracted human teeth of adult patients: molars and premolars removed for prosthetic and periodontitis, with naturally exposed root cement to a depth of 6 mm, i.e., the limit above which the treatment of choice is invasive, surgical treatment. The teeth with a preserved anatomical crown and root were included in the study, whereas the teeth with a loss of root cement during extraction were excluded. The teeth were kept in a prepared 0.5% chloramine solution until the start of the study. Before the beginning of the study, the teeth were cleaned of sediment and calculus, as well as hard and soft tissue debris with use of dental curettes (LM-Instruments, Parainem, Finland). The teeth were then polished with SuperPolish paste (Kerr Dental, Bioggio, Italy), with an RDA of 9.8 and without fluoride, with an Opti Shine toothbrush (Kerr, Bioggio, Italy) using 2000 r/min. The teeth were rinsed three times with distilled water and left for 24 h. 

### 2.1. Demineralization of Tooth Root Cement

A solution was prepared to demineralise the hard tissues of the tooth, the composition of which is shown in Table 1 [25,38].

The teeth were divided into two groups. Then, 1 L of demineralisation solution was poured into one container in which 6 teeth (the test group) were placed. pH = 6.2 of the solution was then determined using acetic acid and potassium hydroxide, depending on the fluctuation of the pH value [39]. The container was placed in an incubator at 37 °C for a period of 5 weeks. The conditions in the incubator mimicked the conditions in the oral cavity to initiate the process of initial caries formation. During the entire 5-week period, the pH was measured daily using a CP 411 pH-meter (Elmetron, Gliwice, Poland), with adjustments of high pH values with acetic acid and low pH values with potassium hydroxide to maintain a constant level of pH = 6. The remaining 6 teeth, which constituted the control group, were kept in distilled water. After 5 weeks, the teeth of both the test and control groups were rinsed three times with distilled water and thoroughly dried.

### 2.2. Preparation of the Experimental Solution

In cooperation with the Department of Organic Chemistry, Bioorganic Chemistry and Biotechnology of the Silesian University of Technology (Gliwice, Poland), an experimental solution was prepared, which consisted of the components listed in Table 2 [40].

An experimental solution with the characteristics of a dental infiltrant was prepared with a mass of 5 g, and it was placed in a dark bottle and mixed thoroughly. DMAEMA in the preparation is the component responsible for polymerisation. PMMAn was enriched with metronidazole to obtain the antimicrobial component. In order to facilitate observation under the microscope and to be able to assess the penetration of the preparations into the dental hard tissue, a dye called eosin (WarChem, Warsaw, Poland) (0.25 mL) was added to both experimental and Icon preparations, as shown in Figure 1. 

### 2.3. Tooth Infiltration Procedure

Two zones were delineated on the surface of both decalcified teeth (study group) and non-decalcified teeth (control group). Each zone was marked with colour varnish on the surfaces of tooth crowns. The blue zone marked teeth infiltrated with the Icon preparation containing eosin, and the red zone marked teeth infiltrated with the experimental preparation with eosin, as shown in Figure 2.

Both preparations were applied according to the instructions for use supplied with the commercial preparation.

During polymerisation, a C01-C Premium Plus wireless LED polymerisation lamp (Premium Plus International Limited, Bournemouth, UK) was used, emitting light in the wavelength range of 440–480 nm. The full mode was used for the study, with the radiation power of 1200 mW/cm^2^.

### 2.4. Preparation of Samples for Microscopic Analysis

The teeth were cut along the long axis, perpendicularly to the traced zones. Tooth surfaces were polished with R100 felt discs (Stoddard, London, UK) at 4000 r/min with SeptoDiamond fluoride-free diamond paste (Septodont, Warsaw, Poland). The teeth were then cold-fused in the Metalogis Opti-Mix two-component methylmethacrylate-based resin, (Metalogis s.c., Warsaw, Poland). They were then cut longitudinally, using a Buehler precision cutter (Buehler Holding A.G., Uzwil, Switzerland), after which the cross-sections were made. The cross-sectional surfaces were sanded with waterproof sandpaper of decreasing gradations: P320, P500, P800, and P1000 Metalogis DEMPAX (Metalogis s.c, Warsaw, Poland); and then polished with Struers DP Suspension diamond slurries: 9 μm, 3 μm, 1 μm, 0.25 μm (Struers A/S, Ballerup, Denmark).

### 2.5. Microscope Observations

Observations of longitudinal sections of teeth were made with a Hitachi S-3400N scanning electron microscope (Hitachi Ltd., Tokyo, Japan) and with an Olympus GX71 light microscope (Olympus, Tokyo, Japan) at 50× magnification.

## 3. Results

The first stage of microscope observation was to determine the anatomical structures visible in the Hitachi S-34000N scanning electron microscope, which are shown in Figure 3.

With the use of a scanning microscope, the thickness of root cement in the cervical region was determined to be 7.47 µm, as shown in Figure 4. It was also possible to see the expansion of root cement, with an increasing thickness to 55.6 µm towards the root apex, as shown in Figure 5.

The second stage of microscopic observation was to analyse the degree of penetration of the experimental preparation and the Icon preparation. Both the resin-embedded teeth of the study group and the control group were observed under an Olympus GX71 light microscope (Tokyo, Japan) at 50× magnification. The results of the observations are shown in Figure 6 and Figure 7.

The light microscope observations revealed penetration of both the experimental preparation and the commercial preparation Icon deep into the decalcified hard tissue in the area of the root. The study showed that both preparations penetrated not only the decalcified root cement but that it also reached the root dentine.

Observation under a light microscope made it possible to assess the degree of penetration of the experimental preparation and the commercial preparation Icon in decalcified teeth. In both cases, penetration of the infiltrant into root cement is a desirable phenomenon, as these preparations should be able to penetrate demineralised tooth root tissues.

## 4. Discussion

Contemporary dentistry is faced with a challenge to maintain oral health by preserving as much healthy tooth tissue as possible, to treat caries and periodontal disease early, and to improve the function of the masticatory apparatus in elderly patients. Based on US statistical data from 2020, it is estimated that the group of people aged 65 and over constitutes 17% of the population [41]. Statistical data of European countries, e.g., Poland, also show that the group of people aged 65 and over is steadily growing. The participation of this age group in the total population (old age index) in 2020 was 18.6%, compared to 1990, when it was 10% [42]. Konopka et al. in their study noted that, in recent years, as life expectancy increases in the studied age groups of the population, the number of retained dentition increases [43]. While considering the aging process of the population, it should be noted that geriatric dentistry will constitute a significant percentage of cases covered by prevention and treatment. With age, a number of physiological changes occur, which in case of the dentition are usually atrophic in nature. In many cases, the dental or periodontal age of a patient does not correspond to their biological age. The coexistence of systemic diseases, the deterioration of oral hygiene associated with reduced manual dexterity and poorer vision, as well as past or active periodontal diseases significantly increase the likelihood of developing dental root caries [44,45]. There are also a number of other behavioural and sociodemographic factors that contribute to the occurrence of root caries [46]. Moreover, it is important to mention that, with age, the salivary glands become fibrotic and fatty, which causes a reduction in saliva secretion in the oral cavity and the development of caries [47]. Senile bone atrophy of the maxillary alveolar process and the alveolar part of the mandible together with gingival recession affects apparent elongation of the clinical crowns and exposure of root cement. Barczak et al. in their study estimated that the incidence of periodontal disease in the elderly occurs in 98–100% of patients [27]. Katz RV et al., while evaluating teeth with recessions, proved that as many as one in nine cases had root caries (11.4%) [32]. Moreover, Slavkin noted that more than 50% of the people in the age group of 65 and older had a diagnosis of root caries [48]. Hellyer et al. determined that the majority of active root carious lesions were located within 1 mm of the gingival margin, while inactive lesions were located above or 1 mm from the gingival margin [49]. It should be mentioned that not only elderly patients but also adolescents who undergo orthodontic treatment, especially with brackets located in the cervical region of teeth and at the same time on a high-sugar diet, are at risk of caries in the cervical region and in the root cement [47].

Until recently, the treatment of primary carious lesions was based on two basic mechanisms: prophylactic and curative, i.e., the use of an appropriate diet and fluoride prophylaxis—the effect of exogenous fluoride compounds (tooth brushing, rubbing), as well as the action of endogenous compounds (their oral intake). These measures were applied before any signs of tooth decay occurred, in order to prevent demineralization, or after a white spot was observed. Scribante et al., in their study, confirmed the correctness of enamel remineralization with hydroxyapatite and sodium fluoride, which was also after teeth whitening [50]. The exogenous use of fluoride compounds is aimed at remineralization and is therefore a form of treatment. The gap between preventive and interventional (“surgical”) treatment has been bridged by micro-invasive dentistry with its promising treatment of early carious lesions, based on the infiltration technique [8].

Most studies involving the process of root tissue infiltration have assessed the degree of microhardness. Zhou et al., in their study, evaluated the effect of infiltration on root caries induced by *Streptococcus mutans* biofilm. The researchers demonstrated that the microhardness of the tooth surface and resistance to caries increased significantly after the application of Icon infiltrant [51]. Yazkan and Gurdogan et al. in their study confirmed that the application of Icon infiltrant increased the microhardness of the enamel surface [52,53]. Additionally, Gurdogan et al. in their study found that Icon increased surface roughness, when compared to the control group [53]. Zhou et al. also noted that Icon resin causes an increase in surface roughness compared to a decalcified surface without application of the infiltrant. That research may suggest that Icon resin application tends to increase plaque accumulation on the tooth surface at the application site [51]. M-Skucha Nowak et al. in their previous study pointed out that the commercial preparation Icon does not meet one of the main characteristics of a dental infiltrant, namely, it does not have a component responsible for bacteriostaticity [54]. In previous studies, the authors demonstrated that the inclusion of metronidazole in the experimental formulation allows for targeted activity against both anaerobic and oxygen-deficient organisms. The PMMAn—MTZ component demonstrates strong antimicrobial activity against *Bacteroides*, *Fusobacterium*, *Eubacterium*, *Clostridium*, *Peptococcus*, *Peptostreptococcus*, and protozoa [26]. Both the authors and a researcher, Zhang, assessed that the inclusion of an antibacterial component in the composition of a dental infiltrant could potentially inhibit the development of caries at the site of application [26,55]. Clinically, this is of great importance because the exposed surface of root cement provides a site of plaque retention. The authors, in their previous study, also demonstrated that an experimental formulation with a PMMAn-MTZ component at the 2-fold dilution applied to an early carious lesion in close proximity to the gingival mucosa showed similar cytotoxicity to the commercial formulation Icon after 24 h, while, at the 4-fold dilution, the experimental formulation showed less cytotoxicity than Icon, suggesting that application of the experimental formulation does not require isolation of the treatment field [26]. Neres et al. focused on the abrasion resistance of Icon on the enamel surface by subjecting it to 10,000 brushing cycles, proving that when the application of the preparation is performed according to the manufacturer’s recommendations in the form of two layers of infiltrant, the preparation will not be abraded [56]. These results allow us to conclude that the application of infiltrants to the exposed root surface subjected to regular hygienic procedures will allow the preparation to persist and protect against the development of caries.

Numerous researchers have infiltrated the enamel surface. Skucha-Nowak et al., while applying the experimental preparation on the enamel surface, proved that the preparation penetrates the pores to a depth of 30–40 µm. The study obtained values similar to those described by Subramaniam et al. [25,57]. The authors of the present study attempted to evaluate the ability of the experimental preparation to penetrate deep into the root cement. By evaluating the degree of penetration of the experimental preparation and the commercial preparation Icon in decalcified teeth, the degree of penetration in both groups was observed, which is a desirable phenomenon, as these preparations should penetrate the demineralised tissues of the root. The phenomenon of infiltration shows that preparation reaches into pores or cavities on the tooth surface. The literature reports that infiltration to a depth of 60 µm is sufficient to prevent demineralisation of enamel tissues [58]. It is known that, depending on the type of tooth, thickness of the enamel can vary from 2.6 mm on the chewing surface to 2.4 mm in molars and premolars, respectively. The thinnest layer was found in the cervical region of the tooth root and was 0.1 mm. In comparison, the thickness of root cement was 150–200 micrometres in the periapical region, while, in the cervical region, it was about 20–50 micrometres. Based on these values, it can be estimated that it is sufficient for the infiltrant to penetrate to a depth of 2.3% of enamel thickness. The authors of this study performed a pioneering study based on a subjective exclusion in the SEM image. In the future, the use of X-ray microtomography (µ-CT) will allow clinicians to accurately analyse the depth of penetration of the experimental preparation into hard tissues [59,60]. This method can provide valuable information in a non-invasive manner, without cutting the samples. These studies need to be continued in order to thoroughly assess the penetration of the experimental preparation based on a statistical study. The results obtained in the study allow for a preliminary assessment that resin infiltration can be used not only on the enamel surface but also within the root. Conservative treatment of elderly patients should be carefully planned. Routine diagnostics combined with treatment of early carious lesions on the root surface may hinder caries progression, resulting in improved oral health by preserving the maximum number of teeth in that group of patients.

The obtained microscopic results showed that infiltrants not only penetrate decalcified root cement but also penetrate the surface area of root dentine.

## 5. Conclusions

The pioneering, preliminary study based on microscopic viewing has proven that the experimental preparation and the features of a dental infiltrate enriched with an antimicrobial component in the form of metronidazole can potentially be used to treat early carious lesions within the tooth root. This study proved that it penetrates into demineralized tissues. There is a need to continue research.

## Figures and Tables

**Figure 1 materials-14-05654-f001:**
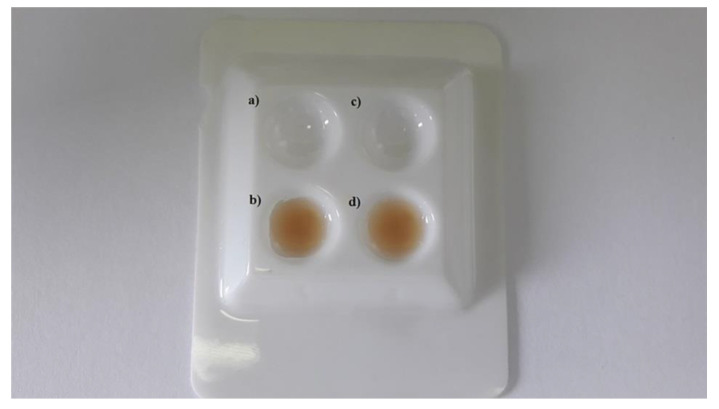
Observation of change in colour of the polymerised experimental preparation (**a**) before the addition of eosin and (**b**) after the addition of eosin, as well as of Icon solution (**c**) before the addition of eosin and (**d**) after the addition of eosin.

**Figure 2 materials-14-05654-f002:**
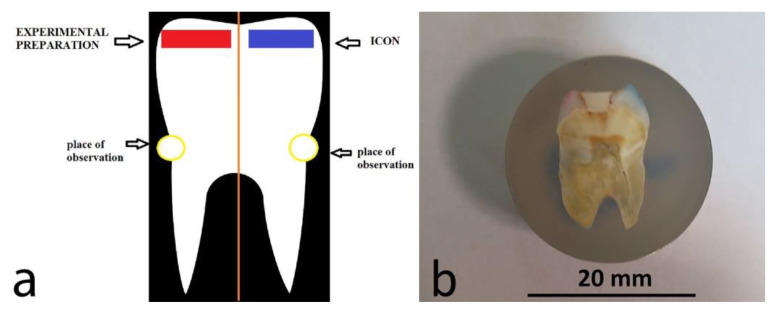
(**a**) Graphically marked tooth zones and areas for microscopic observation; (**b**) a photo of a resin-embedded tooth imaged with a stereoscopic microscope. The red side shows the zone with the applied experimental preparation, and the blue side shows the tooth with the applied commercial preparation Icon.

**Figure 3 materials-14-05654-f003:**
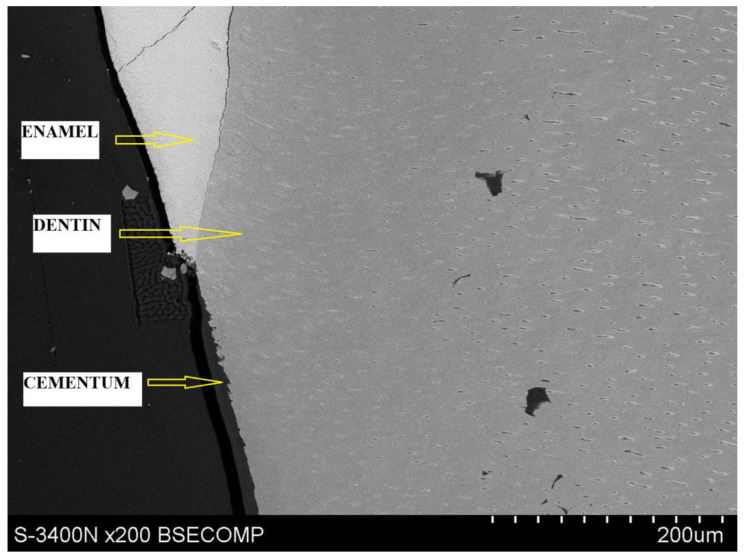
A photograph showing anatomical tooth structures.

**Figure 4 materials-14-05654-f004:**
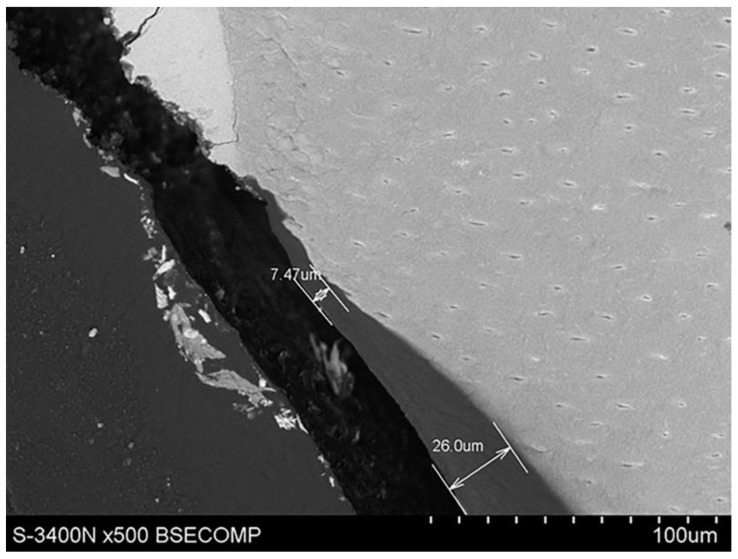
Scanning electron microscope image showing the thickness of root cement in the cervical region.

**Figure 5 materials-14-05654-f005:**
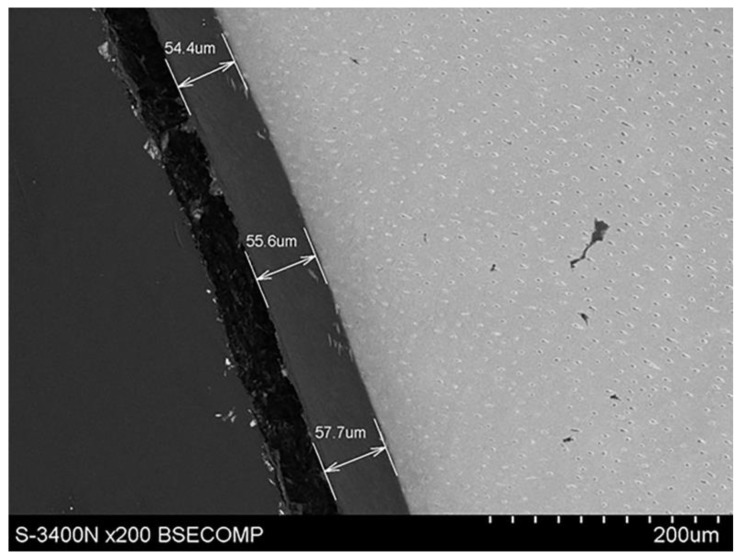
Scanning electron microscope image showing the thickness of root cement below the cervical region, in the apical direction.

**Figure 6 materials-14-05654-f006:**
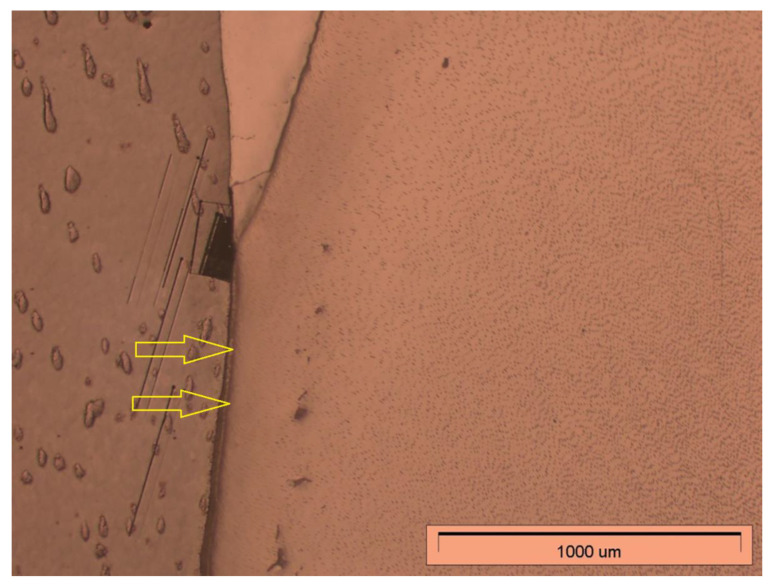
Visible penetration of the experimental preparation in the cervical area of a tooth. The arrows indicate the direction of penetration.

**Figure 7 materials-14-05654-f007:**
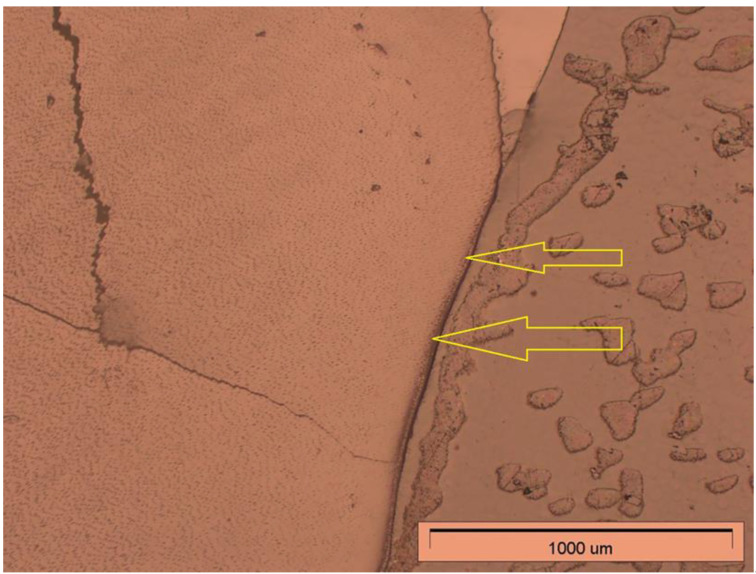
Visible penetration of Icon in the cervical region. The arrows indicate the direction of penetration.

**Table 1 materials-14-05654-t001:** Composition of the solution for demineralization of hard tooth tissues.

Component	Molar Concentration	Quantity
CaCl2*2H2O	3 mM	0.441 g
KH2PO4	3 mM	0.408 g
CH3COOH	50 mM	2.88 mL
MHDP	6 µM	1.416 × 10^−3^ g

**Table 2 materials-14-05654-t002:** Composition and percentage content of the experimental solution.

Component	Quantity (g)	Content (%)
TEGDMA	3.75	75
HEMA	1.25	25
PMMAn-MTZ ^1^	0.05	1 *
DMAEMA ^2^	0.05	1 *
CQ ^3^	0.025	0.5 *

* ratio to total mass of monomers; TEGDMA (triethylene glycol dimethacrylate, Fluka, Buchs, Switzerland); HEMA (2-hydroxyethyl methacrylate, Acros, New Jersey, USA); ^1^ PMMAn (2-(7-methyl-1,6-dioxo-2,5-dioxa-7-octenyl) trimellitic anhydride); MTZ (metronidazole, Acros, New Jersey, USA); ^2^ DMAEMA (N,N-dimethylaminoethyl methacrylate, Merck, Darmstadt, Germany); ^3^ CQ (camphorquinone, Aldrich, St. Louis, Mo, USA).

## Data Availability

Data is contained within the article.

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
