# Peer review of "Assessment of the Potential Ability to Penetrate into the Hard Tissues of the Root of an Experimental Preparation with the Characteristics of a Dental Infiltratant, Enriched with an Antimicrobial Component—Preliminary Study"

_materials, 2021, doi:10.3390/ma14195654_

Round 1

Reviewer 1 Report

The present article is entitled as evaluation of the penetration ability into the root hard tissues of an experimental preparation with the characteristics of a dental infiltrant, enriched with an antimicrobial component.

I do not agree with the present form to be published and I have some comments to the authors:

  1. How was the sample size calculation preformed?
  2. The main goal of the article is not clear. Please state them to correlate the hypothesis to the discussion section.
  3. Introduction and discussion section are too long. 
  4. Authors compare the present dye with antimicrobial properties vs Icon as a positive control. Right? What are the present advantages? Are statistical differences among them? How were they obtained?
  5. From lines 145-153: I recommend to re-write with soundness and clear scientific language. 
  6. Line 157: the exposed root cement were naturally made or were made by authors? Area was calculated with ImageJ?
  7. Line 198: eosin brand name? As well as the other products used? Please insert the missing informartion. 
  8. Why do not make Fig 2 with images from Fig2 and Fig3 side by side? It would be beneficial for the readers. 
  9. The discussion section is from lines 273 to 382, but the present results are only shown in text lines (371-381). Please correct this disproportion.
    1. If in the conclusion section this is affirmed: "there is a need for continued research to clarify the methodology for preparing the human tooth root surface for the infiltration process" what is the main advantage of the present study? 

Author Response

Dear Reviewer 1,

Reviewer 2 Report

Dear Authors, thank you for your manuscript entitled "Evaluation of the penetration ability into the root hard tissues of an experimental preparation with the characteristics of a dental infiltrant, enriched with an antimicrobial component.".

I found your article very interesting and I think it deserves to be considered for publication in Materials.

I just have few comments to improve the quality of your work:

  • I think the introduction is too much long. Please resume it
  • Please remove the colors from the tables in the materials and methods section. 
  • I think it could be interesting to discuss the possible use of the experimental preparation with antimicrobial component event for the treatment of the bleached enamel. Please refer to the following article:

Scribante A, Poggio C, Gallo S, Riva P, Cuocci A, Carbone M, Arciola CR, Colombo M. In Vitro Re-Hardening of Bleached Enamel Using Mineralizing Pastes: Toward Preventing Bacterial Colonization. Materials (Basel). 2020 Feb 11;13(4):818. doi: 10.3390/ma13040818. PMID: 32054090; PMCID: PMC7079603.

Thank you again for your submission.

Yours faithfully

Author Response

Dear Reviewer 2,

I would like to thank you for your valuable comments and criticism regarding our article.

Thank you for giving us an very interesting proposal for a research paper that enriched our manuscript.

As sugessted by the Reviewer, the manuscript has been modified.  All changes to the manuscript are marked in red.

Point 1: I think the introduction is too much long. Please resume it.

Response 1: Thank you for note to this aspect. As suggested by the Reviewer, the abstract has been shortened.

Point 2: Please remove the colors from the tables in the materials and methods section.

Response 2: Thank you for your attention. As requested by the Reviewer, the colors in the tables in the Materials and methods section have been removed.

Point 3: I think it could be interesting to discuss the possible use of the experimental preparation with antimicrobial component event for the treatment of the bleached enamel. Please refer to the following article:

Scribante A, Poggio C, Gallo S, Riva P, Cuocci A, Carbone M, Arciola CR, Colombo M. In Vitro Re-Hardening of Bleached Enamel Using Mineralizing Pastes: Toward Preventing Bacterial Colonization. Materials (Basel). 2020 Feb 11;13(4):818. doi: 10.3390/ma13040818. PMID: 32054090; PMCID: PMC7079603.

Response 3: Thank you for your attention in this regard. Our results, which we want to publish in such a reputable journal, present the assessment of the penetration capacity of an experimental preparation with the characteristics of a dental infiltrate, enriched with an antimicrobial component, into the hard tissues of the root, so to enrich the discussion, we added the article proposed by the Reviewer, referring to it in the discussion, and including a reference to it in the literature.

Reviewer 3 Report

The current research aims to evaluate the penetration ability of an experimental infiltrant with antimicrobial properties into hard root tissues compared to a commercially available infiltrant.

The authors formulate two research hypotheses. However, there are no statistical methods used to test these hypotheses. Furthermore, the authors do not provide numerical data to prove the “deep penetration”; there are only subjective observations base on SEM images. The authors are encouraged to include accurate data and statistical tests to support their hypotheses or consider the study as just an observational study, removing the null hypotheses and specifically stating this aspect in the title and throughout the manuscript.

Author Response

Dear Reviewer 3,

Round 2

Reviewer 3 Report

Thank you for the revisions.

Author Response

Dear Reviewer 3,

I would like to thank You for all Your valuable comments and advice that allowed us to improve and make our manuscript better.